# Multilocus Sequence Typing of *Aggregatibacter actinomycetemcomitans* Competently Depicts the Population Structure of the Species

Signe Nedergaard,[a] Anne B. Jensen,[a,b] Dorte Haubek,[b] Niels Nørskov-Lauritsen[a,c]

[a]Department of Clinical Microbiology, Aarhus University Hospital, Aarhus, Denmark
[b]Department of Dentistry and Oral Health, Aarhus University, Aarhus, Denmark
[c]Department of Clinical Microbiology, Odense University Hospital, Odense, Denmark

**ABSTRACT** We developed a multilocus sequence typing scheme (MLST) for *Aggregatibacter actinomycetemcomitans* based on seven housekeeping genes, *adk*, *atpG*, *frdB*, *mdh*, *pgi*, *recA*, and *zwf*. A total of 188 strains of seven serotypes were separated into 57 sequence types. Whole-genome sequences were available for 140 strains, and in contrast to comparison of 16S rRNA genes, phylogenetic analysis of concatenated MLST gene fragments was in accordance with the population structure revealed by alignment of 785 core genes. MLST could not decisively identify the so-called JP2 clone associated with rapidly progressing periodontitis in adolescents, but noticeable clustering of JP2 genotype strains was revealed. The MLST scheme of *A. actinomycetemcomitans* can be assessed at www.pubmlst.org.

**IMPORTANCE** Accurate diagnosis of infectious disease comprise identification, typing, and antimicrobial resistance of the infective agent. Bacteria are sometimes grouped within their species according to expression of specific toxins or particular antimicrobial resistance traits, but explicit typing for infection control and survey of pathogenesis necessitates genetic analysis such as multilocus sequence typing (MLST). Schemes for the most prevalent human pathogens have been available for more than 10 years, and time has come to extend the scrutiny to second-line infectious agents. One such pathogen is *Aggregatibacter actinomycetemcomitans*, which is commonly involved in periodontitis, and more rarely as the cause of infective endocarditis or spontaneous brain abscess. A MLST scheme for *A. actinomycetemcomitans* is now available at www.pubmlst.org. Whole-genome sequencing of a large number of isolates confirms that MLST competently depicts the population structure of the species.

**KEYWORDS** clonal types, endocarditis, leukotoxin promoter region, multilocus sequence analysis, periodontitis, serotype, population genetic analysis, genetic diversity, leukotoxin (LtxA), polymorphism

**A**ggregatibacter actinomycetemcomitans is a Gram-negative, fastidious, rod-shaped bacterium that is part of the human oral microbiota. It was originally cultured together with *Actinomyces* from actinomycotic lesions of humans but has attracted attention due to its association with invasive and oral diseases (1, 2); in 1996, *A. actinomycetemcomitans* was officially designated an etiological agent of periodontitis (3). Moreover, *A. actinomycetemcomitans* is a member of the HACEK group (*Haemophilus*, *Aggregatibacter*, *Cardiobacterium*, *Eikenella*, and *Kingella*) that is responsible for 1.4% to 3% of reported cases of infective endocarditis (2). *A. actinomycetemcomitans* produces leukotoxin (LtxA) and other proteins involved in aggregation, colonization, and invasion (4). LtxA allows the bacterium to evade immune response by inducing apoptosis in white blood cells through a cellular integrin receptor (5). Strains characterized by a

Address correspondence to Niels Nørskov-Lauritsen, niels.noerskov-lauritsen@rsyd.dk.

530-bp deletion in the promoter of the *ltxCABD* operon secrete increased levels of LtxA and are designated the JP2 genotype of *A. actinomycetemcomitans* (6–8).

Surface antigens are commonly used for initial typing of bacteria, and seven serotypes (a through g) of *A. actinomycetemcomitans* have been characterized (1, 9, 10). The specificity is defined by antigenically distinct O-polysaccharide components of the lipopolysaccharide molecules, and typing is frequently performed by PCR. Serotypes a, b, and c are globally dominant, serotypes d and e are rare, and the prevalence of the most recently identified serotypes, f and g, are unknown (11, 12). The population structure of the species was initially addressed by multilocus enzyme electrophoresis (13, 14) and has been detailed by phylogenetic analysis of WGSs, which suggested a separation into five clades entitled according to the included serotypes (15). The position of the outgroup clade designated e' was also conspicuous by multilocus enzyme electrophoresis (ET division VI) (13) and by 16S rRNA sequence (type V) (16). Recently, we proposed dividing the species into three lineages, where lineage I encompasses serotypes b and c, lineage II serotypes a, d, e, f, and g, and lineage III a subset of serotype a strains (17). A separate lineage was not devised for clade e' because of the disputed speciation of such strains (18). Although WGSs have become available at a reasonable cost, multilocus sequence typing (MLST) is still the preferred designation system for initial typing and characterization. A six-gene MLST scheme was proposed (19), but that scheme did not procure general acceptance and the profiles were not organized in a public MLST website. To achieve the benefits of a portable typing scheme, we expanded the work of Haubek and coworkers to organize a classic seven-gene MLST typing scheme (19).

## RESULTS

**Assignment of STs.** The serotype distribution (a to g) of 188 strains was 25, 112, 23, 9, 10, 8, and 1, respectively. Sixty-four of 112 serotype b strains belonged to the JP2 clone, as defined by a specific deletion in the LtxA gene promoter region. Sequence analysis of the seven loci revealed 13 to 18 allele types per locus (*adk* 13 allele types, *atpG* 18 allele types, *frdB* 14 allele types, *mdh* 13 allele types, *pgi* 17 allele types, *recA* 14 allele types, *zwf* 17 allele types). Allele typing of the 188 strains identified 57 different sequence types (STs), numbered according to frequency of detection. Twenty-one STs included more than one isolate, while 36 STs were represented by only a single isolate (ST22 to ST57). Isolates of the same ST were always of identical serotype. ST1 included 68 serotype b strains. ST2 comprised 17 serotype a strains, ST3 comprised 14 serotype b strains, ST4 comprised seven serotype d strains, and ST5 comprised six serotype c strains. ST6 and ST7 each encompassed four strains of serotypes b and c, respectively. ST8 to ST11 were represented by three strains of serotypes e, b, f, and e, respectively. Sixty-four JP2 clone strains were distributed in seven STs (see below).

**Clonal complexes.** The cluster analysis grouped 51 STs into five clonal complexes (CCs) and left six STs as singletons (Fig. 1). CC1 consisted of 33 STs of serotype b and c, with ST1 as the founding genotype. CC2 included seven STs of serotypes a, d, and g, with ST2 as the founding genotype. CC3 encompassed all five STs of serotype f, with ST50 as founding genotype. CC4 included all four STs of the disputed clade e', with ST11 as the founding genotype. CC5 encompassed two STs representing two isolates of serotype a. Six STs of serotypes a, b, or e were singletons.

**Sequence variation within loci.** The seven-gene fragments of the MLST scheme encompassed 3,497 nucleotides, of which 332 (9%) were variable. All variations between alleles were caused by point mutations, and the number of polymorphic sites of each locus ranged from 30 for the *mdh* allele to 57 for the *zwf* allele. The degree of selective pressure to each locus was determined by calculating the ratio of synonymous to nonsynonymous mutations. The ratio ranged from 0.0026 for *recA* to 0.1600 for *mdh*, implying purifying selection on all of the genes. The degree of recombination between alleles at the different loci was estimated statistically based on the allelic profile data using the classical index of association, $I_A$ (20). The $I_A$ value is expected to be equal to zero if there is linkage equilibrium, meaning that a given population is freely recombining. The value of the index of association

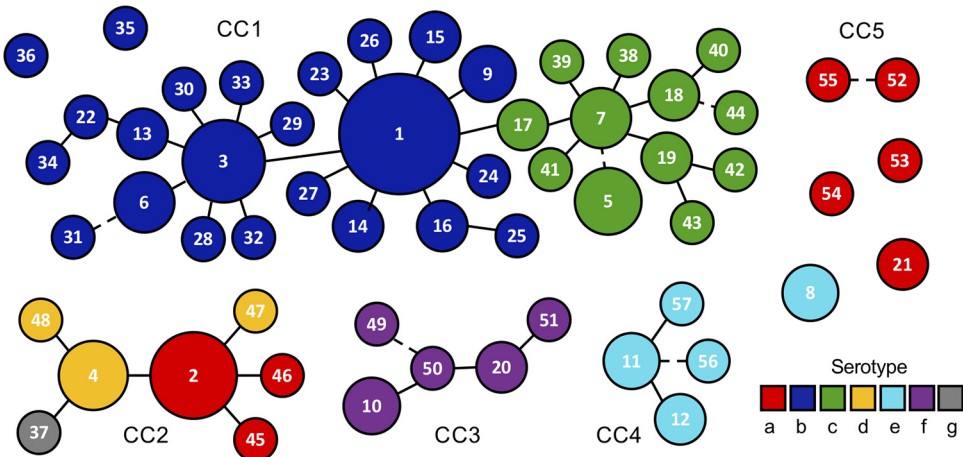

**FIG 1** Clustering of *A. actinomycetemcomitans* sequence types. A total of 188 strains are assembled in 57 STs that are connected by solid or dotted lines (signifying one or two diverging alleles, respectively). The size of the circles displays the number of strains within individual STs (1 to 68). Singleton STs have fewer than 5 alleles in common with any other ST. The color code of serotypes is indicated.

was 2.32 and thus significantly different from zero, indicating linkage disequilibrium and implying clonality and limited recombination within the population.

**Phylogenetic analyses.** WGSs were available from representatives of 55 STs but not from ST16 and ST24. To calibrate the phylogenetic precision of the gene fragments used for MLST, one representative of the 55 STs was selected randomly, and dendrograms were constructed based on 785 core genes, concatenated MLST allele sequences, and 16s rRNA gene fragments (Fig. 2A to C). The population structure disclosed by MLST was similar to the structure revealed by alignment of 785 core genes, particularly with respect to the formation of three evolutionary lineages of the species. Inconsistencies were observed mainly within lineage I, where the MLST dendrogram created two separate clusters of serotype b representatives; remarkably, this separation was almost congruent with the JP2 clone classification, the exception being the non-JP2 strain ST26 that was positioned in the JP2 subcluster. Moreover, the striking singularity of ST44 (of serotype c) by MLST sequence was supported only partially by WGS. This strain shares one allele type (*frdB* allele type 6) with all seven strains of the clade e' outgroup and thus has likely been subjected to recombination of the *frdB* gene. The comparison of 16S rRNA gene fragments recognized the disputed speciation of clade e' strains but was unable to delineate the lineages of the species (Fig. 2C).

The designation of clonal complexes reflected the overall population structure of the species. The seven clade e' outgroup strains all belonged to CC4, and with the exception of two STs categorized as singletons, CC1 encompassed all strains of lineage I. Lineage II strains are presently distributed in CC2 and CC3 (plus singleton ST8), but typing of additional strains may merge strains of this lineage into a single clonal complex. Only six strains of lineage III have been described, and the population structure of this lineage is currently unknown.

**The JP2 clone.** Sixty-four JP2 clone isolates belonged to seven STs (ST1, ST9, ST14, ST15, ST16, ST23, and ST25). With the exception of the large ST1, where 15 of 68 isolates were non-JP2 strains, the 6 other STs were composed solely of JP2 genotype strains (1 to 3 each). Concatenated core gene sequences of all available WGSs of serotype b strains were compared (65 strains of 19 STs, including 24 JP2 genotype strain of six STs) (Fig. 3A). Three large clusters were observed by alignment of 785 genes, encompassing 33, 18, and 7 strains, respectively. The largest group contained all strains of the JP2 clone, plus nine additional strains (group 1, Fig. 3A); JP2 genotype strains formed a subcluster within the group, which was supported by a bootstrap value of 100%. Serotype b strains of ST3 were positioned in two separate clusters, disclosing the superior typing capacity of WGS analysis. The two MLST singletons of serotype b

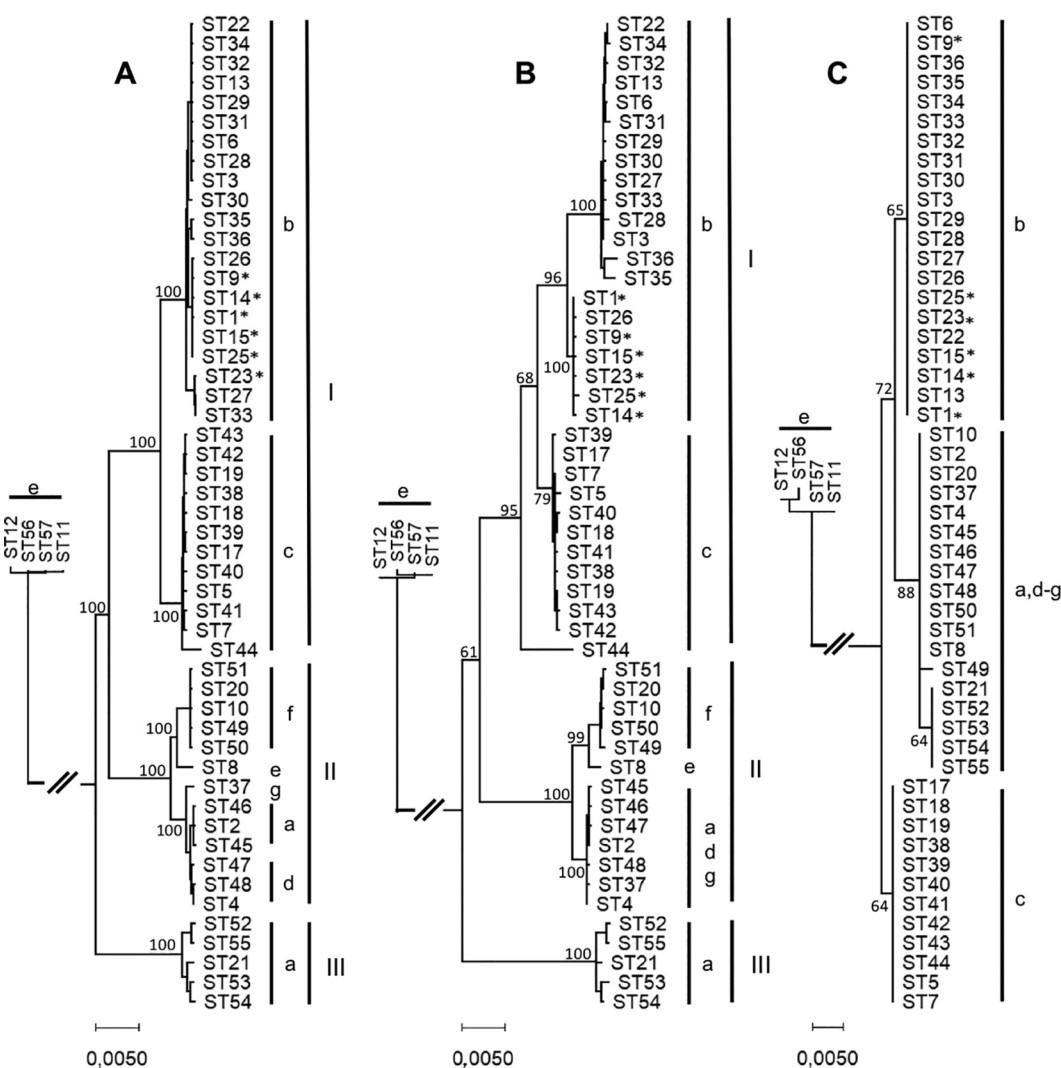

**FIG 2** Neighbor-joining trees with representatives of each of 55 STs. (A) Dendrogram based on 785 concatenated core genes (total length 718,391 bp), (B) dendrogram based on 7 concatenated MLST alleles (total length 3,497 bp), (C) dendrogram based on fragments of 16S rRNA gene (500 bp). Serotypes and phylogenetic lineages are shown. Strains of the JP2 clone are marked with an asterisk (*). The unit of the bars is the number of base substitutions per site. Numbers next to branches are the bootstrap percentage values of 1,000 replicates.

(ST35 and ST36) were also rather unique by comparison of WGS (Fig. 3A). Twenty-eight strains of *A. actinomycetemcomitans* cultured from bloodstream infections that were identified during an examination of Danish HACEK cases of bacteremia (17, 21) were also included in this study. All bloodstream isolates of serotype b clustered together in the second group in Fig. 3A. The third group in the figure includes seven strains isolated from individuals of Ghanaian descent (Fig. 3A, group 3). In contrast, concatenated MLST sequences separated only two clusters of serotype b and were unable to separate the Ghanaian strains from the Danish bloodstream isolates (Fig. 3B).

## DISCUSSION

MLST has proven to be a robust tool in the study of epidemiology and population structures of various species of bacteria (22). The aim of this study was to create an MLST scheme for *A. actinomycetemcomitans* that can provide an initial discrimination of isolates. Fragments of seven housekeeping genes, *adk*, *atpG*, *frdB*, *mdh*, *pgi*, *recA*, and *zwf*, were selected, and 188 isolates were distributed in 57 STs based on 332 polymorphic sites. The d$N$/d$S$ ratios for each of the seven loci were all less than one, a

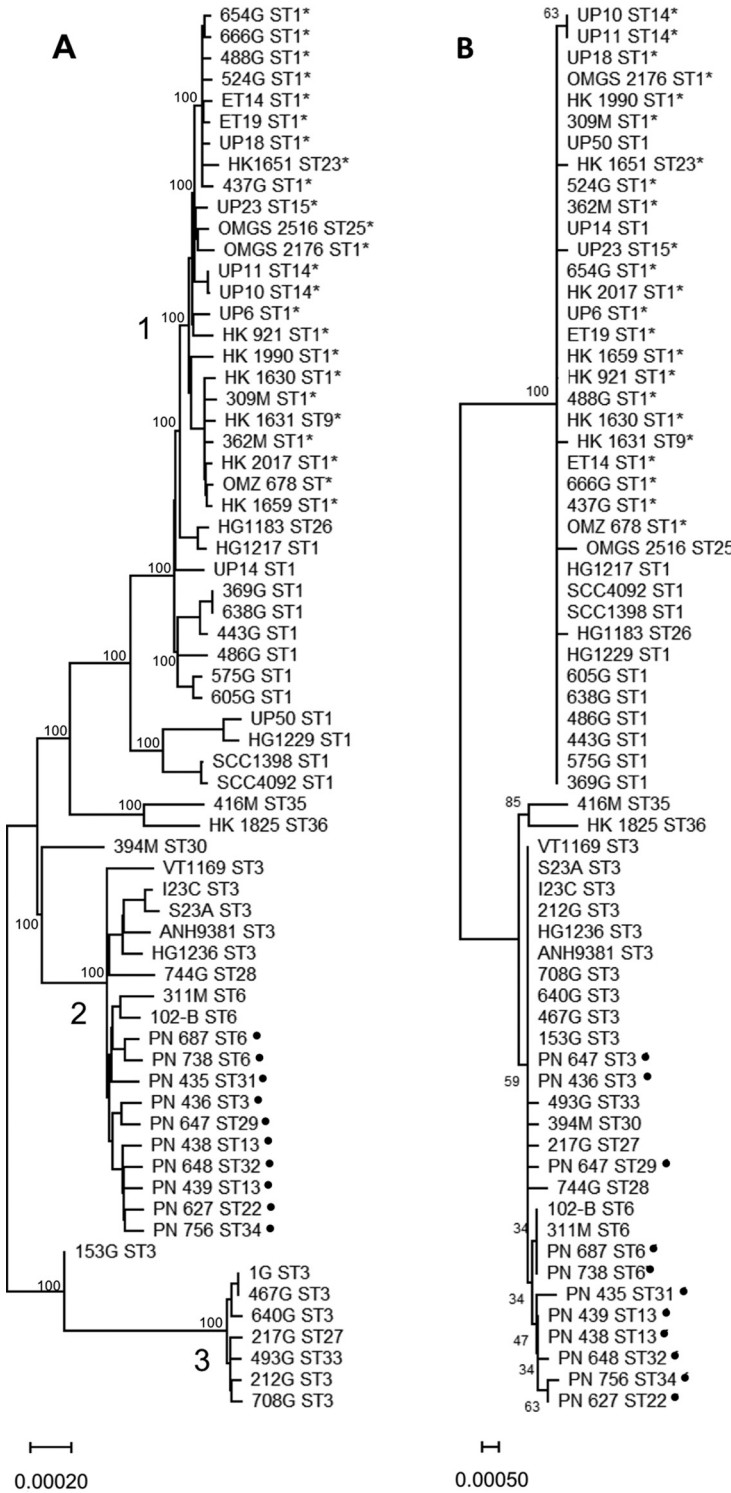

**FIG 3** Neighbor-joining tree with 66 serotype b strains from 19 STs. (A) Dendrogram based on 785 concatenated core genes (718,391 bp total), (B) dendrogram based on 7 concatenated MLST alleles (3,497 bp). Strains of the JP2 clone are marked with an asterisk (*). Strains from cases of bacteremia are marked with a filled-in circle (●). The unit of the bar is the number of base substitutions per site. Bootstrap values (1,000 replicates) are shown next to branches.

finding in accordance with selective neutrality. Fifty-one of 57 STs coalesced into five CCs, leaving six STs as singletons. With the exception of two serotype b singletons, CC1 comprised all serotype b and c strains corresponding to lineage l. The large overrepresentation of serotype b strains (112/188 strains) in this study and in the public

databases springs from attention to the JP2 serotype b clone, which is associated with an aggressive form of periodontitis affecting primarily adolescents (7).

The MLST scheme demonstrated high discriminatory power, as the phylogenetic tree based on concatenated loci (Fig. 2B) resembled the WGS tree (Fig. 2A) in the overall composition and displayed the same discrimination of lineages and clades. In the MLST tree (Fig. 2B), lineage I was separated into three subgroups, as opposed to only two subgroups in the WGS tree (Fig. 2A). A limitation of the MLST scheme was evident by the segregation of ST3 strains into two of the main groups of serotype b as defined by WGS (Fig. 3A, groups 2 and 3). One group consisted solely of ST3 strains isolated from individuals of Ghanaian descent (Fig. 3A, group 3).

As the 16S rRNA gene is ubiquitous and highly conserved, it is often used for initial identification and classification of prokaryotes by Sanger sequencing. In the 16S rRNA tree (Fig. 2C), the phylogenetic lineages were not retained; lineage I was divided into two groups separating serotype b and c strains, and lineage II clustered with lineage III. The 16S tree was unable to reproduce the phylogeny observed in the other trees, stressing the relevance of a MLST scheme as initial classification.

The discriminatory power of MLST distinguished between some isolates of the JP2 clone. Six STs (ST9, ST14, ST15, ST16, ST23, and ST25) comprised solely JP2 strains, but the large ST1 contained both JP2 and non-JP2 strains, and thus MLST cannot identify strains of the JP2 clone with certainty. This also became evident from Fig. 3, as Fig. 3B did not distinguish JP2 from non-JP2 strains. However, all 24 JP2 strains formed a WGS cluster (group 1) in Fig. 3A supported by a bootstrap value of 100%, supporting the clonal structure. Additional JP2-like strains of serotype b with a different (640 bp) deletion in the *ltxCABD* promoter have been reported (23). Whether such strains are part of the JP2 clone must be addressed by MLST and WGS.

The MLST scheme for *A. actinomycetemcomitans* with allele sequences, allelic profiles, and isolate information is publicly available at www.pubmlst.org. Laboratories globally are able to compare their data, and the scheme should provide new insight into the genetic diversity and overall taxonomy of an important human pathogen involved in periodontitis and infective endocarditis.

## MATERIALS AND METHODS

**Selection of alleles, PCR, and Sanger sequencing.** A total of 188 strains were included in the analysis, which were mainly isolated from the oral cavity or from invasive infections of humans. WGSs were available for 140 of these, while 7 genes were sequenced for the remaining 48. Six housekeeping genes (*adk*, *atpG*, *frdB*, *mdh*, *pgi*, and *recA*) that stem from analysis of *Haemophilus influenzae* (24) were previously tested on 82 strains of *A. actinomycetemcomitans* (19). Of the 82 isolates analyzed in that study, 30 have later been genome sequenced and 4 strains were lost. We expanded that scheme with *zwf*, which has been used in MLST analyses of *Mannheimia haemolytica* (25) and *Pasteurella multocida* (26). The *zwf* primers (772fw: ATCAACGCCAATTCCATGCG, 1294rev: CTATTTGCGCGTACCGATGC) were able to amplify fragments of the expected size from 232 isolates of serotypes a to f included in a previous investigation (27) and were used to accomplish a seven-gene MLST scheme for 48 strains. Reactions were performed in 25 $\mu$L volumes with AmpliTaq Gold 360 master mix (Thermo Fisher Inc., Waltham, MA, USA) using denaturation for 5 min at 94°C followed by 30 cycles of 94°C for 1 min, 60°C for 1 min, and 72°C for 2 min and final elongation for 10 min at 72°C. PCR products were treated with ExoSAP-IT Express (Thermo Fisher Inc.) and sequenced on an Applied Biosystems PRISM 3500XL genetic analyzer (Thermo Fisher Inc.) using the PCR primers and the BigDye Terminator v3.1 cycle sequencing kit (Thermo Fisher Inc.). PCR amplification of the other six alleles has been described previously (19). Serotyping was performed by PCR as described previously (27).

**Generation of WGSs and extraction of alleles.** Sixty-six strains of *A. actinomycetemcomitans* were genome sequenced as described previously (17), and WGSs from 74 additional isolates were retrieved from GenBank. The seven alleles were extracted using the online server PATRIC (Pathosystems Resource Integration Center) (28). WGSs of four database sequences were excluded from the analysis: three strains (ANH9776, RHAA1, A160) because of incomplete gene fragments and one strain (Y4, accession no. AMEN00000000.1) because Y4 is a reference strain of serotype b but the sequence clustered within clade a/d, suggesting an erroneously labeled sequence.

**Allele calling, sequence type assignment, and cluster analysis.** For each of the seven loci, unique alleles were assigned an allele number, and the seven allele numbers of individual strains constitute the allelic profile. Unique allelic profiles were assigned a sequence type (ST). A clonal complex (CC) was defined as a group of STs that differed at only one or two alleles compared to at least one other ST of the complex, while STs that varied at three or more loci relative to all other STs were designated singletons. Within each CC, the founding genotype was defined as the ST with the greatest number of single locus variants (29). Clustering of related STs into CCs was performed in PHYLOViZ 2.0 with default settings for clustering (30).

**Data analysis and phylogeny.** Using concatenated allele sequences (total length: 3,497 nucleotides [nt]) in the order *adk*, *atpG*, *frdB*, *mdh*, *pgi*, *recA*, and *zwf*, phylogenetic trees were constructed using the neighbor-joining algorithm in MEGA X (31). From genome-sequenced strains, 785 core genes were extracted (718,391 nt), concatenated, and aligned as described previously (17), and 500 bp 16S rRNA gene fragments (nt 812 to 1,311, covering variable regions 5 to 8) were retrieved from the WGSs using PATRIC. Phylogenetic analysis was conducted in MEGA X. The ratio between numbers of synonymous and nonsynonymous mutations (d$N$/d$S$), the index of association ($I_A$), and the number of polymorphic sites were evaluated with START2 software (32).

**Data availability.** Seventy-four whole-genome sequences (WGSs) were downloaded from NCBI and included in the study. Sixty-six additional strains were genome sequenced and deposited in the NCBI genome database under BioProject accession number PRJNA633763. To add on to a previous characterization, *zwf* allele sequences of 48 strains have been deposited to NCBI under accession numbers MW017158 to MW017203.

## ACKNOWLEDGMENTS

We thank Mogens Kilian, Aarhus University Denmark, and Anders Johansson and Rolf Claesson, Umea University Sweden, for provision of bacterial strains and discussion of periodontal phenotypes.

N.N.L. conceived the project. N.N.L. and S.N. designed the project. S.N. and A.B.J. performed PCR and sequencing. S.N. analyzed data. S.N. made the first draft of the manuscript, which was revised critically for important intellectual and clinical content by N.N.L., A.B.J., and D.H. All authors contributed to the manuscript and approved the final version. This work received no specific grant from any funding agency.

We declare no conflicts of interest.

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
