## [Reviewer comments · Microbiology Spectrum]

Microbiology Spectrum

Multilocus sequence typing of *Aggregatibacter actinomycetemcomitans* competently depicts the population structure of the species

Signe Nedergaard, Anne Jensen, Dorte Haubek, and Niels Nørskov-Lauritsen

Corresponding Author(s): Niels Nørskov-Lauritsen, Odense University Hospital

Review Timeline:

Submission Date:	July 30, 2021
Editorial Decision:	August 23, 2021
Revision Received:	October 22, 2021
Accepted:	October 25, 2021

Editor: S. Wesley Long

Reviewer(s): Disclosure of reviewer identity is with reference to reviewer comments included in decision letter(s). The following individuals involved in review of your submission have agreed to reveal their identity: Tomonori Hoshino (Reviewer #1)

Transaction Report:

DOI: <https://doi.org/10.1128/Spectrum.01085-21>

August 23, 2021

Prof. Niels Nørskov-Lauritsen
Odense University Hospital
Clinical Microbiology
Odense C
Denmark

Re: Spectrum01085-21 (Multilocus sequence typing of *Aggregatibacter actinomycetemcomitans* competently depicts the population structure of the species)

Dear Prof. Niels Nørskov-Lauritsen:

Thank you for submitting your manuscript to Microbiology Spectrum. When submitting the revised version of your paper, please provide (1) point-by-point responses to the issues raised by the reviewers as file type "Response to Reviewers," not in your cover letter, and (2) a PDF file that indicates the changes from the original submission (by highlighting or underlining the changes) as file type "Marked Up Manuscript - For Review Only". Please use this link to submit your revised manuscript - we strongly recommend that you submit your paper within the next 60 days or reach out to me. Detailed information on submitting your revised paper are below.

Link Not Available

Sincerely,

S. Wesley Long

Journals Department
Reviewer comments:

Reviewer #1 (Comments for the Author):

Major comments

Author tried to show the MLST analysis based on seven housekeeping genes, *adk*, *atpG*, *frdB*, *mdh*, *pgi*, *recA*, and *zwf*, and/or 785 concatenated core genes had the advantage in the study of epidemiology and population structures of *Aggregatibacter actinomycetemcomitans*. In Figure 2, I think that the MLST analysis could extract the clade associated with periodontal disease pathogenic population, JP2 strains. In this point, I agree with the opinion of the author and this article is meaningful in the field of clinical periodontology. On the other hand, in Figure 3, why is not the NJ phylogenetic tree based on seven housekeeping genes provided? I think that the NJ tree based on 785 concatenated core genes could well divided the populations associated with diseases, such as periodontitis and bacteremia. If the NJ tree based on seven housekeeping genes could also divided those populations, it would be more meaningful as the rapid and reliable diagnostic equipment. Thus, I request that the NJ tree based on seven housekeeping genes is provided in Figure 3.

Minor comments

1. Page 4, line 4

HACEK group should be explained more, although the reference is cited.

2. Page 4, line 51-53

This sentence is complicated. Please rewrite it.

3. Page 5, line 74-75

The reference of Haubek's work should be cited.

4. Page 5, line 91-91

Now, "Applied Biosystems" is one brand of Thermo Fisher Inc., Waltham, MA, USA.

5. Page 5, line 91-Page 6, line93

"30 cycles of 94{degree sign}C-1 min/60{degree sign}C-1 min/72{degree sign}C-2 min" is incomprehensible. "30 cycles of 94{degree sign}C for 1 min, 60{degree sign}C for 1 min and 72{degree sign}C for 2 min" or "30 cycles of 1 min at 94{degree sign}C, 1 min at 60{degree sign}C and 1 min at 72{degree sign}C" seems to be comprehensible.

6. Page 7, line 138

What do the numbers e. g. adk "13" mean? Please explain them in detail, although author wrote "per locus".

7. Page 8, line 161

What does "from 30 (mdh) to 57 (zwf) mean? Please provide more information.

8. Page 9, line 163

Are "0.0026" and "0.1600" the ratio of synonymous to non-synonymous mutation?

9. Page 9, line 168

Is (2.32) the classical index of association?

10. Page 10, line194-198

This sentence is too long and complicated. Please check.

Reviewer #2 (Comments for the Author):

This manuscript describes the development of a seven allele MLST typing scheme for *Aggregatibacter actinomycetemcomitans* and its application to elucidate the population structure of this organism. The MLST scheme is based on previous research that established a six allele protocol and its novelty is based on the addition of a seventh allele and the application of the new scheme. There are two main issues with the manuscript.

1. On line 89 the authors state their zwf primers "were able to amplify fragments of the expected size from 232 isolates of serotypes a to f included in a previous investigation [23]...". The referenced study included 257 isolates. The authors need to make clear whether they amplified the gene from 232 out of 232 isolates or if their method failed to amplify the gene from any isolates. If only 232 samples were tested why were these samples selected? If the primers failed to amplify zwf from any members of the species its use in an MLST protocol is in doubt and would need to be defended.
2. The justification for developing the new MLST protocol is that the previous one only used alleles from six housekeeping genes and was not widely accepted. Whereas the use of seven alleles is common there is no absolute requirement for this number. The difference in resolving the population structure of *A. actinomycetemcomitans* using the different schemes, for example as shown in Figure 1, should be compared and included.

Minor points:

The exact zwf allele used for MLST analysis should be identified. Presumably it would be trimmed to not include primer sites.

The genome location of the zwf gene relative to the other genes used in the MLST scheme should be provided.

Staff Comments:

Preparing Revision Guidelines

Please return the manuscript within 60 days; if you cannot complete the modification within this time period, please contact me. If you do not wish to modify the manuscript and prefer to submit it to another journal, please notify me of your decision immediately so that the manuscript may be formally withdrawn from consideration by Microbiology Spectrum.

If you would like to submit an image for consideration as the Featured Image for an issue, please contact Spectrum staff.

This manuscript describes the development of a seven allele MLST typing scheme for *Aggregatibacter actinomycetemcomitans* and its application to elucidate the population structure of this organism. The MLST scheme is based on previous research that established a six allele protocol and its novelty is based on the addition of a seventh allele and the application of the new scheme. There are two main issues with the manuscript.

1. On line 89 the authors state their *zwf* primers “were able to amplify fragments of the expected size from 232 isolates of serotypes a to f included in a previous investigation [23]...”. The referenced study included 257 isolates. The authors need to make clear whether they amplified the gene from 232 out of 232 isolates or if their method failed to amplify the gene from any isolates. If only 232 samples were tested why were these samples selected? If the primers failed to amplify *zwf* from any members of the species its use in an MLST protocol is in doubt and would need to be defended.
2. The justification for developing the new MLST protocol is that the previous one only used alleles from six housekeeping genes and was not widely accepted. Whereas the use of seven alleles is common there is no absolute requirement for this number. The difference in resolving the population structure of *A. actinomycetemcomitans* using the different schemes, for example as shown in Figure 1, should be compared and included.

Minor points:

The exact *zwf* allele used for MLST analysis should be identified. Presumably it would be trimmed to not include primer sites.

The genome location of the *zwf* gene relative to the other genes used in the MLST scheme should be provided.

Reviewer #1 (Comments for the Author):

Major comments

Author tried to show the MLST analysis based on seven housekeeping genes, *adk*, *atpG*, *frdB*, *mdh*, *pgi*, *recA*, and *zwf*, and/or 785 concatenated core genes had the advantage in the study of epidemiology and population structures of *Aggregatibacter actinomycetemcomitans*. In Figure 2, I think that the MLST analysis could extract the clade associated with periodontal disease pathogenic population, JP2 strains. In this point, I agree with the opinion of the author and this article is meaningful in the field of clinical periodontology. On the other hand, in Figure 3, why is not the NJ phylogenetic tree based on seven housekeeping genes provided? I think that the NJ tree based on 785 concatenated core genes could well divided the populations associated with diseases, such as periodontitis and bacteremia. If the NJ tree based on seven housekeeping genes could also divided those populations, it would be more meaningful as the rapid and reliable diagnostic equipment. Thus, I request that the NJ tree based on seven housekeeping genes is provided in Figure 3.

Reply: Figure 3 has been updated to include a NJ tree based on seven housekeeping genes.

Minor comments

1. Page 4, line 4

HACEK group should be explained more, although the reference is cited.

Reply: [Line 48-49] Genera of the HACEK group are now mentioned.

2. Page 4, line 51-53

This sentence is complicated. Please rewrite it.

Reply: [Line 50-53] The sentence has been rewritten.

3. Page 5, line 74-75

The reference of Haubek's work should be cited.

Reply: [Line 72] The citation has been added

4. Page 5, line 91-91

Now, "Applied Biosystems" is one brand of Thermo Fisher Inc., Waltham, MA, USA.

Reply: [Line 91] This has been corrected.

5. Page 5, line 91-Page 6, line93

"30 cycles of 94{degree sign}C-1 min/60{degree sign}C-1 min/72{degree sign}C-2 min" is incomprehensible. "30 cycles of 94{degree sign}C for 1 min, 60{degree sign}C for 1 min and 72{degree sign}C for 2 min" or "30 cycles of 1 min at 94{degree sign}C, 1 min at 60{degree sign}C and 1 min at 72{degree sign}C" seems to be comprehensible.

Reply: [Line 92] This has been rephrased.

6. Page 7, line 138

What do the numbers e. g. adk "13" mean? Please explain them in detail, although author wrote "per locus".

Reply: [Line 134-137] This has been specified.

7. Page 8, line 161

What does "from 30 (mdh) to 57 (zwf) mean? Please provide more information.

Reply: [Line 158-159] This has been specified.

8. Page 9, line 163

Are "0.0026" and "0.1600" the ratio of synonymous to non-synonymous mutation?

Reply: [Line 160-161] Yes, this has now been specified.

9. Page 9, line 168

Is (2.32) the classical index of association?

Reply: [Line 165] Yes, this has now been specified.

10. Page 10, line 194-198

This sentence is too long and complicated. Please check.

Reply: [Line 194-199] The sentence has been adjusted.

Reviewer #2 (Comments for the Author):

This manuscript describes the development of a seven allele MLST typing scheme for *Aggregatibacter actinomycetemcomitans* and its application to elucidate the population structure of this organism. The MLST scheme is based on previous research that established a six allele protocol and its novelty is based on the addition of a seventh allele and the application of the new scheme. There are two main issues with the manuscript.

1. On line 89 the authors state their *zwf* primers "were able to amplify fragments of the expected size from 232 isolates of serotypes a to f included in a previous investigation [23]...". The referenced study included 257 isolates. The authors need to make clear whether they amplified the gene from 232 out of 232 isolates or if their method failed to amplify the gene from any isolates. If only 232 samples were tested why were these samples selected? If the primers failed to amplify *zwf* from any members of the species its use in an MLST protocol is in doubt and would need to be defended.

Reply: We amplified the gene from 232 out of 232 isolates and thus the method has not failed to amplify the gene from any isolates. We tested 232 samples because the DNA from these 232 isolates had just been purified as part of another project (Jensen et. al 2019) and included isolates from all lineages of the species.

Reference: Jensen, A.B., et al., *Comprehensive antimicrobial susceptibility testing of a large collection of clinical strains of Aggregatibacter actinomycetemcomitans does not identify resistance to amoxicillin*. J Clin Periodontol, 2019. **46**(8): p. 846-854.

2. The justification for developing the new MLST protocol is that the previous one only used alleles from six housekeeping genes and was not widely accepted. Whereas the use of seven alleles is common there is no absolute requirement for this number. The difference in resolving the population structure of *A. actinomycetemcomitans* using the different schemes, for example as shown in Figure 1, should be compared and included.

Reply: The 2007 study only focused on variants from the JP2 clone and is therefore not comparable to this scheme which includes all branches of the species. The scheme was never uploaded to pubmlst. Our goal with this article is therefore to strengthen the work from 2007 and finally make it publicly available. We have not made changes to the 2007 scheme with regard to the six genes, we just updated it and broadened the analysis by adding the seventh gene (*zwf*) and by including much more strains in the analysis. The seventh gene was added in order to live up to the guidelines of a classical MLST scheme. The seventh gene also makes the analysis more robust, as only 51 sequence types were identified when using only the original six genes to analyze the 188 strains.

Minor points:

1. The exact *zwf* allele used for MLST analysis should be identified. Presumably it would be trimmed to not include primer sites.

Reply: The *zwf* allele sequence without primers is available from the pubmlst website (https://pubmlst.org/bigsdb?db=pubmlst_aactinomycetemcomitans_seqdef&page=plugin&name=FastaExport)

2. The genome location of the *zwf* gene relative to the other genes used in the MLST scheme should be provided.

Reply: Two deposited and closed genomes are commonly used as reference, namely HK1651 (positions *adk*; 1269235–1269793, *atpG*; 998161–998660, *frdB*; 551859–552368, *mdh*; 1213887–1214326, *pgi*; 1098608–1099134, *recA*; 1045025–1045519, *zwf*; 800615–801077), and D7s (see Figure below). The housekeeping genes are well separated in both genomes. Additional closed genomes are becoming available, and the positions of the genes vary. We are reluctant to include positions in specific strains in the manuscript.

October 25, 2021

Prof. Niels Nørskov-Lauritsen
Odense University Hospital
Clinical Microbiology
Odense C
Denmark

Re: Spectrum01085-21R1 (Multilocus sequence typing of *Aggregatibacter actinomycetemcomitans* competently depicts the population structure of the species)

Dear Prof. Niels Nørskov-Lauritsen:

Your manuscript has been accepted, and I am forwarding it to the ASM Journals Department for publication. You will be notified when your proofs are ready to be viewed.

Sincerely,

S. Wesley Long
Editor, Microbiology Spectrum
